# Quantifying Reinforcement-Learning Agent’s Autonomy, Reliance on Memory and Internalisation of the Environment

**DOI:** 10.3390/e24030401

**Published:** 2022-03-13

**Authors:** Anti Ingel, Abdullah Makkeh, Oriol Corcoll, Raul Vicente

**Affiliations:** 1Institute of Computer Science, University of Tartu, Narva mnt 18, 51009 Tartu, Estonia; oriol.corcoll.andreu@ut.ee; 2Göttingen Campus Institute for Dynamics of Biological Networks, University of Göttingen, 37075 Göttingen, Germany

**Keywords:** autonomy, reinforcement learning, information theory, partial information decomposition, non-trivial informational closure, deep learning

## Abstract

Intuitively, the level of autonomy of an agent is related to the degree to which the agent’s goals and behaviour are decoupled from the immediate control by the environment. Here, we capitalise on a recent information-theoretic formulation of autonomy and introduce an algorithm for calculating autonomy in a limiting process of time step approaching infinity. We tackle the question of how the autonomy level of an agent changes during training. In particular, in this work, we use the partial information decomposition (PID) framework to monitor the levels of autonomy and environment internalisation of reinforcement-learning (RL) agents. We performed experiments on two environments: a grid world, in which the agent has to collect food, and a repeating-pattern environment, in which the agent has to learn to imitate a sequence of actions by memorising the sequence. PID also allows us to answer how much the agent relies on its internal memory (versus how much it relies on the observations) when transitioning to its next internal state. The experiments show that specific terms of PID strongly correlate with the obtained reward and with the agent’s behaviour against perturbations in the observations.

## 1. Introduction

Reinforcement learning (RL) is a biologically plausible type of learning in which an agent learns by trial and error while interacting with its environment [1]. Fuelled by deep neural architectures, artificial RL agents can develop long-term strategies to explore and exploit the structure and reward signals in complex environments. Such agents have recently achieved impressive performance in a suite of environments ranging from board and video games to real-world practical problems, including robotics [2,3,4,5]. Intuitively, the agent’s success is explained in terms of a certain adaptation or internalisation by the agent to regularities of the environment, including the environment’s response to the agent’s actions.

The success of RL agents is almost invariably characterised by the amount of reward obtained in a certain amount of episodes or time. After all, the agent’s learning is driven to maximise its cumulative reward, and hence, its reward score is indicative of the success in solving a task. However, a single scalar hardly indicates what the agent has actually learned or internalised. In particular, the same performance can result from agents with different levels of reactivity to the current state of the environment. While some tasks promote agents acting purely reactively, other tasks could induce the emergence of internal states that allow a certain decoupling from the current environmental state.

Moreover, as we will argue, the level of reliance of the agent on the environmental state (as opposed to a higher level of reliance on its internal state) has important implications for its behaviour against different types of perturbations. Hence, a refined characterisation of an RL agent’s adaptation and behaviour needs to go beyond the global reward obtained. In particular, it would need to evaluate how much the agent is being influenced by the environment and its internal states, respectively. Currently, it is not known how the influence of the environment upon an RL agent evolves as the agent progresses through its training. Does it usually increase or decrease during the learning? Does it change monotonically?

At the intuitive level, the autonomy of an agent has been associated with the level of shielding of the agent’s goals and behaviour from the immediate control by the environment in which it is situated. Based on such a notion, information-theoretic measures of autonomy have been formally introduced by Bertschinger et al. [6].

More generally, information theory and its different functionals have historically served to formalise measures about the degree and direction of influence between agents as well as between the agent and the environment [7,8,9,10,11,12,13]. Furthermore, some of these measures have been used for intrinsic reward to guide the agent’s learning [14]. Partial information decomposition (PID) has emerged recently as the information-theoretic tool to decompose the information that a pair of random variables contain about a third one [15]. Its use to characterise and drive the learning of biological and artificial systems is a current direction of interest [16,17,18].

Adaptability and autonomy of artificial agents are considered necessary requirements for the agent to act flexibly and robustly under changing real-world conditions [19]. This line of research has, for example, led to developing self-programming agents [20]. For an agent to be adaptable, it has to be perturbation tolerant [19], for which self-monitoring is a fundamental requirement [21].

In this study, we capitalise on the measures by Bertschinger et al. [6] and PID to characterise the evolution of the autonomy of RL agents over their training. We introduce an algorithm for calculating these and other information-theoretic measures in the limiting process of time step approaching infinity, allowing to monitor these measures. Further, we use information theory to characterise a certain kind of perturbation tolerance. In this work, we have assembled multiple fundamental artificial intelligence problems under a common information-theoretic setting.

In the following, we first present the information-theoretic framework for the measures and introduce an algorithm for calculating the measures in Section 2. Next, we describe the experiments’ setup and results. In particular, we report the level of autonomy as an agent learns and becomes more successful for two different environments. The use of PID allows us to decompose the environmental and internal state influence on the agent’s next state. We also test how the different PID terms predict the robustness of agents’ behaviour to different perturbations. Finally, we discuss the limitations and implications of our results, their relation to previous literature, and future directions.

## 2. Materials and Methods

In the following, we first describe the information-theoretic framework in Section 2.1. This is the framework in which the used measures are defined. Details for calculating the measures are given in Section 2.1.4. Next, we introduce PID generally in Section 2.2 and finally, we discuss the specifics of applying PID to the autonomy measures and non-trivial informational closure (NTIC) in Section 2.3. The algorithm introduced in this work for calculating the information-theoretic measures is available in a code repository https://github.com/antiingel/RL-agent-autonomy (accessed on 15 January 2022).

### 2.1. Information-Theoretic Framework

In this section, we describe the information-theoretic framework in which Bertschinger et al. [6] introduced the measures of autonomy, and we describe our method for calculating the measures introduced by Bertschinger et al. [6]. First, we define the fundamental information-theoretic quantities. Next, we describe the Markovian structure of the system for which the measures are defined, and then we introduce the measures themselves. In the last subsection, we describe the details of calculating the required probabilities.

#### 2.1.1. Preliminaries

In this section, we give an overview of the basic notions of information theory. Throughout this paper, we work with discrete random variables with a finite number of states. Let *X* denote a random variable taking values in some finite set $\left\{x_{1},\ldots ,x_{n}\right\}$. Let us denote the probability measure of the probability space on which *X* is defined as **P**. Then entropy of *X* is defined as: $$H\left(X\right)=-\sum_{i=1}^{n}P(X=x_{i})\log_{2}\left(P(X=x_{i})\right),$$
where $0\log_{2}0$ is defined to be equal to 0. Suppose now that *Y* is another random variable on the same probability space. Then random vector (X,Y) is also a random variable, and we can calculate its entropy H(X,Y). Conditional entropy is defined as: H(X∣Y)=H(X,Y)−H(Y).

Next, we define mutual information between *X* and *Y* as: MI(X:Y)=H(X)−H(X∣Y).

Finally, we define conditional mutual information. For that, suppose we a have third random variable, *Z*, on the same probability space. Conditional mutual information between *X* and *Y*, given *Z*, is: MI(X:Y∣Z)=H(X∣Z)−H(X∣Y,Z).

#### 2.1.2. Markovian Structure

For our experiments, we use the framework by Bertschinger et al. [6], in which an agent interacts with an environment, and the interactions between them are analysed. Thus, we assume that there is a distinction between the agent and the environment. We assume that time evolves in discrete steps. We denote the agent’s state at time step *n* as $S_{n}$ and similarly the environment’s state as $E_{n}$. Similarly to the partially observable Markov decision process (POMDP), the agent cannot directly see the state of the environment in this framework. However, there is a random variable, $O_{n}$, whose distribution is fully determined by the state of the environment $E_{n}$. The agent can use it together with $S_{n}$ to determine its next state $S_{n+1}$. Correspondingly, the agent’s state does not directly affect the environment. However, there is a random variable $M_{n}$, the motor action, which can affect the next environment’s state $E_{n+1}$ and whose distribution is fully determined by the state of the agent $S_{n}$. See Figure 1 for the interactions.

We assume that the sequence ${\left\{\left(S_{n},E_{n}\right)\right\}}_{n=0}^{\infty }$ forms a Markov chain and that $S_{n+1}$ and $E_{n+1}$ are conditionally independent given $S_{n}$ and $E_{n}$. We assume that this Markov chain is homogeneous. Practically, we achieve homogeneity by stopping the learning process and analysing the agent’s policy at that training step. This analysis can be done separately at different time steps. Between those time steps, learning can take place.

#### 2.1.3. Autonomy and Non-Trivial Informational Closure

In this framework, Bertschinger et al. [6] introduce quantitative measures to characterise the agent’s behaviour. They define two different measures of autonomy, suitable for different situations. If the environment drives the agent, which means that $E_{n+1}$ depends only on $E_{n}$ and not on $M_{n}$, Bertschinger et al. [6] define the autonomy measure as: (1)$$A_{m}=\mathrm{MI}(S_{n+1}:S_{n}|E_{n},E_{n-1},\ldots ,E_{n-m}),$$
where *m* is a non-negative integer denoting the length of the sequence of the environment’s considered states. We considered only the case m=0 throughout this work; thus, we denote it $A_{0}$. If the agent drives the environment, which means that $S_{n+1}$ depends only on $S_{n}$ and not on $O_{n}$, Bertschinger et al. [6] define the autonomy measure as: (2)$$A^{*}=\mathrm{MI}(S_{n+1}:S_{n}).$$

Some motivations for these definitions are given in Section 2.3.1, Section 2.3.2 and Section 5.1. For further details, refer to [6]. In addition to $A_{0}$ and $A^{*}$, Bertschinger et al. [6] define: (3)$$\mathrm{NTIC}=\mathrm{MI}(S_{n+1}:E_{n},\ldots ,E_{n-m})-\left(S_{n+1}:E_{n}|S_{n}\right).$$
We considered only the case m=0 for NTIC throughout this work. In case NTIC >0, it shows how much the agent models the correlations in the environment. The other case, NTIC <0, refers to a synergistic situation where the agent’s and environment’s previous states together determine the agent’s next state.

#### 2.1.4. Input to Autonomy Measures and NTIC

Bertschinger et al. [6] introduce multiple interesting information-theoretic measures, such as autonomy and NTIC, in the described framework. To simplify their calculation, we use a stationary distribution of the Markov chain ${\left\{\left(S_{n},E_{n}\right)\right\}}_{n=0}^{\infty }$ or its estimate as an input to measures (Equation 1)–(Equation 3). Using stationary distribution removes the dependence on the time step. The stationary distribution can be interpreted as a limiting distribution for an aperiodic Markov chain.

In more detail, given the transition matrix and the distribution of $\left(S_{0},E_{0}\right)$, one can calculate the fraction of time spent in each state in the long run as: $$\mu \left(s,e\right)=\lim_{n\to \infty }\frac{1}{n}\sum_{i=0}^{n-1}P\left(S_{i}=s,E_{i}=e\right)$$
for each environment’s state *e* and agent’s state *s*. This distribution forms a stationary distribution of the Markov chain.

The probabilities μ(s,e) can be calculated as follows. First, find the communicating classes of the Markov chain. Each communicating class forms an irreducible Markov chain. Calculate the stationary distribution for each communicating class. Since the stationary distribution is unique for irreducible Markov chains, any of the available methods for calculating it can be used. Finally, to obtain the stationary distribution for the whole Markov chain, the stationary distributions for each class have to be merged together and weighted by the probability of reaching the corresponding class.

We used Python (version 3.7.6) package discreteMarkovChain (https://github.com/gvanderheide/discreteMarkovChain, accessed on 15 January 2022) (version 0.22) to calculate the probabilities μ(s,e). According to the package’s documentation, the power method for calculating the probabilities is robust even if there are hundreds of thousands of states.

In order to calculate $A_{0}$, $A^{*}$, and for m=0, the probabilities μ(s,e) are not enough. We need probabilities for triples $\left(s^{\prime },s,e\right)$ where *s* and *e* denote the agent’s and environment’s current states and $s^{\prime }$ denotes the next agent’s state. Thus, we denote the fraction of time spent in each triple $\left(s^{\prime },s,e\right)$ as $\mu (s^{\prime },s,e)$, which can be calculated from the probabilities μ(s,e) and the transition matrix. If the transition matrix is known, we use $\mu (s^{\prime },s,e)$ as the input to $A_{0}$, $A^{*}$, and NTIC.

If the transition matrix is not known, we estimate the fraction of time spent in each state, denoted $\hat{\mu }\left(s^{\prime },s,e\right)$, by simply counting the number of times the state occurred in a sample and dividing it by the number of elements in the sample. We use this as an input distribution to calculate $A_{0}$, $A^{*}$, and NTIC for m=0. Thus, we are using the plug-in method. The distribution $\hat{\mu }\left(s^{\prime },s,e\right)$ (or $\mu (s^{\prime },s,e)$) is also the input to PID calculation algorithms.

### 2.2. Partial Information Decomposition

This section describes PID, which allows for decomposing the autonomy measures and NTIC discussed in Section 2.1.3. This decomposition gives a more refined look at these measures, possibly giving a better characterisation of the agent’s behaviour. This section introduces PID generally for the required number of variables. Later we give the relationships between the measures and PID terms and discuss the decomposition of the autonomy measures.

PID extends classical information theory by making it possible to decompose the mutual information into several components [15]. PID partitions MI$\left(T:S_{1},S_{2}\right)$, the information that a set of source random variables $S_{1}$ and $S_{2}$ have about a target random variable *T*, into different information contributions of the source variables [15]. PID partitions MI$\left(T:S_{1},S_{2}\right)$ into four parts:The unique contribution of $S_{1}$, denoted by UI$\left(S_{1}\right)$, which is the information gained about *T* from accessing $S_{1}$ and cannot be gained otherwise;The unique contribution of $S_{2}$, denoted by UI$\left(S_{2}\right)$, which is the information gained about *T* from accessing $S_{2}$ and cannot be gained otherwise;The synergistic contribution of $S_{1}$ and $S_{2}$, denoted by CI, which is the information gained about *T* from accessing both $S_{1}$ and $S_{2}$ and cannot be gained otherwise;The shared (or redundant) contribution of $S_{1}$ and $S_{2}$, denoted by SI, which is the information gained about *T* when either $S_{1}$ or $S_{2}$ are accessed.

The relationship between the PID terms and classical quantities can be summarised in the following system: (4)$$\begin{array}{cc} \mathrm{MI}(T:S_{1},S_{2}) & =\mathrm{CI}+\mathrm{SI}+\mathrm{UI}(S_{1})+\mathrm{UI}(S_{2}),\\ \mathrm{MI}(T:S_{1}) & =\mathrm{SI}+\mathrm{UI}(S_{1}),\\ \mathrm{MI}(T:S_{2}) & =\mathrm{SI}+\mathrm{UI}(S_{2}). \end{array}$$
These partitionings are visualised in Figure 2. PID can be generalised to more than two sources [15,22], but in this work, we only need to consider the case of two sources (representing the internal state of the agent and the environment’s state). In this case, each PID term together with the basic relations (Equation 4) determines the three other terms, and thus, it is sufficient to estimate one of them. Different methods for estimating PID terms are available, and a unifying PID measure is still missing [15,23,24,25,26,27,28,29,30,31,32,33]. For decomposing autonomy measures and NTIC, we use $\tilde{\mathrm{UI}}$ proposed by Bertschinger et al. [24] and $I_{\cap }^{\mathrm{SX}}$ proposed by Makkeh et al. [33]. See Appendix B for further discussion.

### 2.3. Decomposing Autonomy Measures and NTIC

The autonomy measures and NTIC described in Section 2.1.3 can be decomposed into PID terms. In this case, the target variable is $S_{n+1}$, and the source variables are $S_{n}$ and $E_{n}$. Using Equation (Equation 4), one can derive the following relationships: $$\begin{array}{cc} A_{0} & =\mathrm{CI}+\mathrm{UI}(S_{n}),\\ A^{*} & =\mathrm{SI}+\mathrm{UI}(S_{n}),\\ \mathrm{NTIC}=\mathrm{SI}-\mathrm{CI}. \end{array}$$
Next, we discuss the intuition behind the definitions of the autonomy measures and their decompositions.

#### 2.3.1. Decomposition of $A_{0}$

Bertschinger et al. [6] suggest using $A_{0}=\mathrm{MI}(S_{n+1}:S_{n}|E_{n})$ to quantify autonomy in the scenarios where the environment drives the agent. By the chain rule for mutual information: $$\mathrm{MI}(S_{n+1}:S_{n},E_{n})=\mathrm{MI}(S_{n+1}:E_{n})+\mathrm{MI}(S_{n+1}:S_{n}|E_{n}).$$

Thus, the total mutual information MI$\left(S_{n+1}:S_{n},E_{n}\right)$ can be partitioned into MI$\left(S_{n+1}:E_{n}\right)$, the information gained about $S_{n+1}$ by accessing $E_{n}$, and MI$\left(S_{n+1}:S_{n}|E_{n}\right)$, the information gained about $S_{n+1}$ if accessing $S_{n}$ is required. Different partitionings of mutual information are visualised in Figure 2.

Decomposing $A_{0}$ into unique information UI $\left(S_{n}\right)$ and synergistic information CI allows for a more detailed interpretation of the measure. The measure $A_{0}$ seems to account for autonomy since it measures the information about the future state of the agent gained by accessing the current state either alone (unique information contribution) or in combination with the current environment’s state (synergistic contribution). PID quantifies the amount of these contributions separately.

#### 2.3.2. Decomposition of $A^{*}$

Bertschinger et al. [6] suggest using $A^{*}=\mathrm{MI}(S_{n+1}:S_{n})$ to quantify autonomy in the scenarios where the agent drives the environment. The measure $A^{*}$ shows the information gained about $S_{n+1}$ by accessing $S_{n}$. It reflects how much the agent is in control of its dynamics. We refer to [6] for a more detailed discussion.

The measure $A^{*}$ can be decomposed into unique information UI$\left(S_{n}\right)$ and shared information SI. Shared information shows the coherence of the agent and the environment. High shared information could be interpreted as the agent having more control over the environment.

## 3. Experiment Setup

This section describes the experiments conducted using RL to evaluate the measures of autonomy described in Section 2.1.3. We use two different settings. First, we consider a theoretically tractable case. In this case, we use the policy iteration algorithm to guarantee convergence in the training process. Further, we show that using the stationary distributions discussed in Section 2.1.4 allows us to calculate all the required probabilities and measures. In this environment, the agent affects the environment, in which case $A^{*}$ could be considered to be the suitable autonomy measure. A similar environment was discussed by Bertschinger et al. [6].

In the second case, we consider a more practical situation where the transition probabilities are unknown and must be estimated. The transition probabilities are estimated by a histogram in this case. Thus, we use the plug-in method. We use deep RL for training and the agent’s memory as its internal state in this case. This environment corresponds to the case where the agent is driven by the environment, and, thus, $A_{0}$ is considered to be the suitable autonomy measure.

We monitor the changes in information-theoretic measures in both cases. The first environment demonstrates that it is possible to define an internal state for the agent if one is not readily available through the training method. In the second case, we evaluate if unique information can be used to determine if the agent relies more on its memory or observation. We introduce perturbations as a control to test the agent’s reliance on memory. The following sections give more details about each of the environments.

### 3.1. Grid Environment

In this experiment, the environment is a 5×5 grid. The agent starts in a random location in the grid. At every time step, the agent can move to any adjacent square (left, right, up, or down) or stay at its current location. Food can appear in the grid’s corners, and the agent is rewarded for being in the same location as the food. If the agent is in the same location as the food, the food disappears and later reappears in some corner. In addition to the positive reward of the food, the agent gets a negative reward for each moving action. The food has a probability of disappearing at every time step before the agent reaches the food. The following sections give more details about the environment, the training process, and the agent.

#### 3.1.1. States of the Environment

From the point of view of the information-theoretic framework introduced in Section 2.1, the environment’s state is the food’s location—any of the four corners or a no-food state. Let us denote the probability of the food disappearing at any time step before the agent reaches it as *d*. If the environment is in the no-food state at some time step, then at the next time step, any of the five states are chosen uniformly at random as the next state. If the state is a corner of the grid, then with probability *d*, the next state will be the no-food state, and with probability 1−d, the state remains the same. Let us denote the corner states as $e_{i}$ for i∈{1,2,3,4} and the no-food state as $e_{0}$. The transitions of the environment can be summarised as: $$\begin{array}{cc} P(E_{n+1}=e_{i}|E_{n}=e_{i}) & =1-d\\ P(E_{n+1}=e_{0}|E_{n}=e_{i}) & =d\\ P(E_{n+1}=e_{j}|E_{n}=e_{0}) & =0.2, \end{array}$$
where i∈{1,2,3,4} and j∈{0,1,2,3,4}. See Figure 3a for the transition diagram. In these experiments, the observation $O_{n}$ is equal to the environment’s state $E_{n}$, which means that the environment is fully visible to the agent. Refer to Section 2.1.2 for definitions of $E_{n}$ and $O_{n}$.

#### 3.1.2. Training

We consider the setting as a Markov decision process (MDP) for training the agent. There is also a notion of a state in MDP, but this state does not coincide with the environment’s state or agent’s state of the information-theoretic framework. Thus, there is a third notion of state. We differentiate between these states because we are using the policy iteration algorithm (see [1] Section 4.3) to train the agent, and in this setting, there is no readily available and easily interpretable internal state for the agent. Instead, we define an internal state of the agent using policies obtained from training (see Section 3.1.3).

In MDP, the state must consist of all the information available to the agent. Thus, the states of the MDP are chosen to consist of the food’s location and the agent’s location. The rewards are +10 for being at the same location as the food and −1 for each movement. The food location changes as described in Section 3.1.1, and the agent’s location changes according to its policy. The initial policy of the agent is uniformly random, meaning that in each state, every action has the same probability. The agent is trained using policy iteration.

Policies obtained at each iteration of the policy iteration algorithm (that is, after each policy evaluation) are saved. The threshold for stopping the policy evaluation (denoted by θ in [1] Section 4.3) was chosen to be $10^{-6}$. The saved policies are used to analyse the agent’s behaviour at different time steps. We use the greedy policy, meaning that the action corresponding to the highest value is chosen.

#### 3.1.3. Internal States of the Agent

In this experiment, the agent does not have an internal state that it can directly manipulate. Thus, the state of the agent is not readily available, and it has to be defined. One possibility would be to define the agent’s state as its location. However, our preliminary experiments showed that this approach does not give easily interpretable results since this state has, in a sense, multiple roles: internal state and location.

Therefore, we define a more high-level state for the agent. Namely, we define the state as the corners where the agent is heading. This way, we obtain a state that indicates the agent’s intention and, thus, this state is more suitable to be considered the agent’s internal state. In more detail, the possible states are the subsets of the corners $\left\{e_{1},e_{2},e_{3},e_{4}\right\}$. The subset can contain multiple corners (if there is a positive probability of ending up in different corners) or be empty. Details of finding the new states and calculating their transition probabilities are given in Appendix A. The transitions of the state depend on the policy of the agent. Policies throughout training are obtained as described in Section 3.1.2.

### 3.2. Repeating-Pattern Environment

In this experiment, the environment is a repeating pattern of symbols. In some iterations, the pattern is completely visible to the agent, and in other iterations, the pattern is invisible. Whether the pattern is visible or invisible for a given iteration is decided randomly. The agent has to match its action to the pattern to receive a reward. The agent is provided with a memory to be able to solve this task. This simple setting allows us to analyse how the agent uses its memory and to what extent it internalises the state of the environment. Reliance on memory and environment is tested by applying perturbations to the observations and the pattern. Unlike in the grid environment, here we are using deep RL, which better reflects a practical situation, and we are estimating the required probabilities using a histogram. Thus, we are using the plug-in method. The following sections give more details about the environment, the training process, and the agent.

#### 3.2.1. States of the Environment

The repeating pattern is an ordered triple of symbols, denoted $\left(p_{1},p_{2},p_{3}\right)$, and the pattern is formed from an alphabet of two symbols, {1,2}. In our experiments, the repeating pattern is $\left(p_{1},p_{2},p_{3}\right)=\left(2,1,1\right)$. These values are not the environment’s states. Environment’s states are defined separately to accommodate having visible and invisible states.

In order to have visible and invisible states, six environment states are needed, three visible states and three invisible states. We denote these states $v_{1},v_{2},v_{3}$ and $i_{1},i_{2},i_{3}$, respectively. The possible observations for the agent are {0,1,2}. The environment cycles over the pattern and produces an observation reflecting the current value of the pattern in a visible state or 0 in an invisible state. The process of producing observations can be formalised as a function *O* from states to observations defined as: $$\begin{array}{cc} O(v_{j}) & =p_{j}\\ O(i_{j}) & =0. \end{array}$$

Whether an iteration will be visible or not is decided randomly at the end of the previous iteration. With probability *h*, it is invisible, and with probability 1−h, it is visible, where *h* is a parameter that we control. See Figure 3b for the transition diagram. Different values of *h* can promote the learning of different strategies, i.e., we would expect that the agent relies more on its memory for larger values of *h*.

#### 3.2.2. Training

In order to train an agent in this setting, we consider it a POMDP. The states of the POMDP are the states of the environment $\left\{v_{1},v_{2},v_{2},i_{1},i_{2},i_{3}\right\}$. The set of possible actions for the agent is {1,2}, and the set of observations is {0,1,2}. At a given environment’s state *e*, the environment produces an observation *o* using the function *O*. The agent uses its policy $\pi_{z}$ to select an action *a*, and consequently, the environment provides a reward *r* and the next observation $o^{\prime }$. The policy depends on the agent’s memory state, denoted by *z*.

The agent is given a reward of 1 when the action produced matches the current value of the pattern $p_{j}$. Otherwise, the agent is punished by 0.9. Consequently, the reward is maximised when the agent is able to replicate the pattern with its actions. We would expect this task to be trivial when the pattern is fully visible since the agent just needs to copy its input to its output. Thus, the environment can hide the state by producing an observation 0.

The agent cannot rely on the observations to solve the task when the state is invisible. Thus, we provide it with a memory and consider the state of the memory as the agent’s state. The agent and its memory are implemented as a neural network that approximates the Q-value function $Q_{z}\left(o,a\right)$. The network consists of a linear layer of size 64 with ReLU activation, an LSTM [34] with 32 units as the memory, and a linear layer to map the memory to the set of actions.

The parameters of the neural network are trained using vanilla Deep-Q-Networks (DQN) [35]. Samples of the form $\left(o,z,a,z^{\prime },o^{\prime },r\right)$ were used in the training process. Here, *z* and $z^{\prime }$ denote the memory state in consecutive steps, *o* and $o^{\prime }$ denote the observations in consecutive steps, *a* denotes the action, and *r* denotes the reward. The samples were collected using the policy $\pi_{z}$, which is implemented as: $$\pi_{z}\left(o\right)=\underset{a}{\mathrm{arg max}}\;Q_{z}\left(o,a\right),$$
where *o* denotes an observation, *a* denotes an action, and *z* denotes the memory state. Figure 4 depicts the interactions of the agent and the environment. It is important to notice that the agent can solve the task in various ways. For example, it can rely solely on memory or in a mix of memory and observations, both strategies leading to optimal behaviour.

#### 3.2.3. Internal States of the Agent

As mentioned in the previous section, the agent’s state is taken as the memory’s state, and the memory is implemented as an LSTM. More precisely, since discrete random variables are needed, the agent’s state corresponds to the binned memory value. The agent’s state has 15 possible values, corresponding to 15 equal-width bins. The smallest value and the largest value from the collected data were taken as the start of the first bin and the end of the last bin, respectively.

Data was collected in order to estimate the information-theoretic measures at different training steps. The training process was frozen every 100 training steps to collect data for estimating the required probabilities. The data was collected from 20 episodes, each episode lasting 30 time steps. At each of the 600 time step, the state of the LSTM was saved. Since the memory values were binned, the required probabilities were simply estimated by a histogram. From these probabilities, the information-theoretic measures can be calculated (see Section 2.1.3 for definitions and Section 2.1.4 for calculation details). Thus, we are using the plug-in method to estimate the information-theoretic measures.

#### 3.2.4. Success of the Agent

We compare the information-theoretic measures to the success of the agent. In particular, we define overall success as the fraction of correct actions by the agent over the last 20 episodes; and we define hidden success as the fraction of correct actions when the pattern is not visible, again over the last 20 episodes. Comparing other measures to the overall success and hidden success allows us to see how predictive these measures are of the agent’s success in the original or the perturbed environment.

#### 3.2.5. Perturbations

Usually, the strategy learned by the agent is treated as a black box, and information-theoretic quantities could help to characterise the agent’s behaviour. We introduce two different perturbations into the environment and analyse the agents’ behaviour in the perturbed environments. We test if UI$\left(S_{n}\right)$ calculated in the original environment can predict the success of the agent in perturbed environments.

The observation perturbation aims to evaluate whether the agent relies more on its memory than on observations. In contrast, the pattern perturbation evaluates whether the agent relies more on the observations, which means that the internal memory is used as a passthrough module.

The observation perturbation adds noise to the observation, i.e., with a probability of 0.2, the part of the pattern observed by the agent will be swapped with another value present on the pattern. For example, if the environment would output 2 at a time step, the perturbation will replace it with 1 instead. The perturbation happens only for the visible states. The hypothesis is that if the agent relies on its memory, its performance will not degrade since its memory is robust to noisy observations.

The pattern perturbation replaces the pattern (2,1,1) on which the agent was trained with another pattern (2,2,2). The intuition behind this perturbation is that the agents that are relying on the observation more than the memory would not be affected by it.

## 4. Results

The following subsections give the results of the two experiments described in Section 3.1 and Section 3.2. We calculated the information-theoretic measures to characterise the learning of the agents. We used PID to decompose these measures into more fine-grained terms. The decompositions were obtained using the BROJA estimator [36]. We obtained qualitatively similar results with the SxPID estimator [33]. Refer to Appendix C for examples. The grid environment and repeating-pattern environment results are given in Section 4.1 and Section 4.2, respectively.

### 4.1. Results on Grid Environment

Recall that the grid environment described in Section 3.1 is the theoretically tractable case in which the agent can affect the environment and, thus, $A^{*}$ could be considered the appropriate measure of autonomy. In this setting, transition probabilities are known and do not have to be estimated.

We considered two food disappearing probabilities for the grid environment, d=0.1 and d=0.04. Since the agent receives a negative reward for any movement, having a high probability of food disappearing means it is not beneficial to always chase after the food. An optimal policy is obtained through the training since the policy-iteration algorithm is used [1]. In the case d=0.04, the optimal policy is to always go after the food. In the case d=0.1, the optimal policy is to go after the food only if the food is close enough. The results are presented in Figure 5 and Figure 6.

Figure 5 shows the autonomy measures, NTIC, and their PID decompositions (see Section 2.3) over the policy-iteration training process. The autonomy measures and NTIC are introduced in Section 2.1.3, and PID terms SI, CI, and UI$\left(S_{n}\right)$ are introduced in Section 2.2. For both cases, d=0.1 and d=0.04, at iteration 0, the policy is uniformly random, meaning that each action is chosen uniformly at random. It takes the first four iterations to remove this completely random behaviour from the policy since, after four steps, most of the values have been updated. After that, the agent fine-tunes its behaviour more to the corresponding environment. This shift from the first four iterations to the later phase can also be seen in the figure.

In both cases, we see an increase in NTIC, which would normally indicate that the agent’s internalisation of the environment increases. Here, however, we should recall that the agent’s state is the set of destination corners. Thus, a more likely interpretation is that there is a coherence between the destination corners and the environment’s state, as NTIC is almost equal to SI. This interpretation comes from the definition of SI, since PID is applied using destination corners and environment’s state as the source variables.

Unlike NTIC and SI, we see a decrease in the autonomy measures. This could happen because the optimal policies are more restrictive, while non-optimal behaviour allows for more autonomy. The final value of UI$(S_{n})=0$ is expected for the case d=0.04 since the agent always follows the food with the optimal policy in this case.

In Figure 5, we used deterministic initial states for calculating the stationary distributions (see Section 2.1.4). In case the values of the initial states $E_{0}$ and $S_{0}$ were chosen uniformly at random, the results were as seen in Figure 6. As can be seen, the initial randomness can affect the measures considerably, at least in the first half of the training. However, it is more likely that one is interested in how the behaviour affects the measures and not the initial randomness. Thus, Figure 5 should be a better characterisation of the behaviour.

### 4.2. Results on Repeating-Pattern Environment

Recall that the repeating-pattern environment described in Section 3.2 was the more practical case in which agents are trained using deep RL, and transition probabilities had to be estimated. Since the agent is driven by the environment, $A_{0}$ is considered the appropriate autonomy measure.

For the repeating-pattern environment, we considered 11 values for the hiding probability *h*, ranging from 0 to 1 with a 0.1 interval. For each value of *h*, we trained the agent with 30 different random initialisations. Next, we analysed how the autonomy measures, NTIC, and their PID decompositions changed throughout the training. We also compared the information-theoretic measures obtained at the end of the training for different values of *h*. Finally, we explored how the information-theoretic measures are related to the agent’s success in perturbed environments.

#### 4.2.1. Autonomy and NTIC Throughout Training

First, we looked at how the measures changed throughout the training process. Figure 7 shows NTIC and $A_{0}$, together with their PID decompositions (see Section 2.3), the agent’s success, and the total mutual information MI$\left(S_{n+1}:S_{n},E_{n}\right)$. The PID terms SI, CI and UI$\left(S_{n}\right)$ were introduced in Section 2.2. This figure corresponds to one random initialisation of the agent and to the case h=0.9. Since there are few visible states for h=0.9, solving the task is relatively complex and takes many training steps.

With high hiding probability *h*, the agent cannot rely on the observations to get a high success and has to use its memory to model the dynamics of the environment. In the beginning, the success was around $\frac{2}{3}$, which could be obtained by constantly choosing action 1. This happens since the pattern is (2,1,1), and $\frac{2}{3}$ of the symbols are equal to 1. As the training progressed, the agent’s success got close to the perfect score of 1. Thus, we see a correlation between NTIC (which is almost equal to SI since CI is close to zero) and success.

Figure 8 shows scatterplots between overall success and SI over the training. It includes data of all 30 random initialisations of the agent. We see that the higher the value of *h*, the more correlated SI and success become. For small *h*, the agent usually does not have to use its memory since the environment is mostly visible. For large *h*, using memory is needed to have high success. If the agent’s actions are dictated by its internal state $S_{n}$, then the more coherent its internal state is with the environment, the more successful the agent is. Thus, SI will correlate with the agent’s success when the agent has a high success and uses its internal state to calculate its action.

#### 4.2.2. Autonomy and NTIC at the End of Training

In this section, we compare the final values of the measures for different values of *h*. For each random initialisation, we calculate the average values of the measures over the last 10 saved training steps (62,400; 62,500; …; 63,300). The averages and confidence intervals of these average values are shown in Figure 9.

In scenarios where the environment observations were abundant (h≤0.5), the agent had less incentive to internalise the dynamics of the environment. In this case, $A_{0}$ (which is almost equal to UI $\left(S_{n}\right)$) constitutes a small portion of MI$\left(S_{n+1}:S_{n},E_{n}\right)$, and the agent could be considered as not possessing a high level of autonomy. However, when the environment observations were scarce (h>0.5), the agent had to internalise the dynamics of the environment in order to receive a high reward. In such scenarios, $A_{0}$ constituted a dominant portion of MI$\left(S_{n+1}:S_{n},E_{n}\right)$, and the agent could be considered to have a higher degree of autonomy.

The normalised values in Figure 9 are the original values divided by the total mutual information MI$\left(S_{n+1}:S_{n},E_{n}\right)$. It is easier to compare how much each term constitutes to the total mutual information by using normalised values. While the non-normalised $A_{0}$ decreased, the normalised $A_{0}$ still increased, as seen from the figure. The decrease was due to a decrease in the total mutual information. We can see that normalised $A_{0}$ increased almost monotonically over the values of *h*.

#### 4.2.3. Agent in Perturbed Environments

These experiments analyse the strategies learned by different agents, and we test if UI$\left(S_{n}\right)$ is predictive of the agent’s success in perturbed environments. First, we consider the observation perturbation experiments. Figure 10 shows scatterplots between the success in the perturbed environment and the unique information UI$\left(S_{n}\right)$ in the original environment throughout the training process. Unique information UI$\left(S_{n}\right)$ quantifies the reliance of $S_{n+1}$ on $S_{n}$. Again, if *h* is small, the agent does not need to rely on its memory most of the time. However, for large *h*, reliance on memory is required more. We see that the unique information UI$\left(S_{n}\right)$ in the original environment and success in the perturbed environment are more correlated for higher values of *h*.

For pattern perturbations we did not obtain interesting results. In most cases, agents perfectly imitated the perturbed pattern in cases where the pattern was visible, and executed the original pattern in cases where the pattern was invisible. In this case, as expected, PID terms did not have interesting relations to the success in the perturbed environment.

## 5. Discussion

### 5.1. Brief Synthesis of Results

The correlation between shared information and success (Section 4.2.1), and the correlation between unique information and success in the perturbed environment (Section 4.2.3) give empirical evidence that the measures can be applied in practice to monitor the strategy of the agent, as these results coincide with our intuitive expectations about the measures. This also shows the suitability of the introduced algorithm for calculating the measures. In more complex environments, monitoring these measures could help us to understand if the agent is learning even if progress is not seen in the obtained reward.

The results and the theoretical analysis suggest that when the agent affects the environment (grid environment experiments), autonomy manifests in the shared information between the internal state and the environment. Whereas, when the agent is solely driven by the environment (repeating-pattern experiments), autonomy could manifest in the unique information of the internal state.

Recall that $A^{*}=\mathrm{SI}+\mathrm{UI}(S_{n})$ and $A_{0}=\mathrm{CI}+\mathrm{UI}(S_{n})$, thus $A^{*}$ and $A_{0}$ can be viewed as more coarse-grained autonomy measures in the corresponding situations. Fluctuations in synergy can render $A_{0}$ hard to interpret (see Section 5.2.1). Thus, in more complex environments that also exhibit synergy fluctuations, PID is an important tool to devise more fine-grained measures.

Let us now return to the question of how autonomy increases during training. Figure 5 and Figure 7 show that the autonomy measures have an increasing trend throughout most of the training in these simple environments. In the end, there is a decrease in $A^{*}$ and $A_{0}$, which can be explained by the restrictions posed by the optimal behaviour that the agent is close to achieving. However, in scenarios where the agent drives its environment, it is theoretically unclear why the unique information UI$\left(S_{n}\right)$ would evolve monotonically during training, be it increasing or decreasing. Thus more complex environments could exhibit different changes in the autonomy measure $A^{*}$ during training.

### 5.2. Implications

In this work, PID is used as a monitor to get a more refined look into the autonomy measures suggested by Bertschinger et al. [6] when applied to RL. Herein, we discuss some direct implications of PID that scratch the surface beyond monitoring such autonomy measures. We start by presenting how PID averts a possible synergy dilemma in $A_{0}$, and then explain further the robustness of shared information compared to $A^{*}$.

#### 5.2.1. PID Averts the Synergy Dilemma in $A_{0}$

When the environment drives the agent, Bertschinger et al. [6] suggested $A_{0}=\mathrm{MI}(S_{n+1}:S_{n}|E_{n})$ as a measure of autonomy. Using PID, we see that $A_{0}$ decomposes into the unique information of the agent UI$\left(S_{n}\right)$ and the synergistic information of the agent and the environment.

Bertschinger et al. [6] pondered upon whether synergistic information reflects the autonomy of the agent or not. To explain the reason for such uncertainty, we recall that the synergistic information is the information about $S_{n+1}$ that can be retrieved only if $S_{n}$ and $E_{n}$ are simultaneously accessed. Good intuition for synergistic information is illustrated in the XOR gate, where none of the inputs $X_{1},X_{2}$ of the gate can reveal any information about the output *Y* (MI$(Y:X_{i})=0$). However, jointly, they reveal one bit about *Y* (MI$(Y:X_{1},X_{2})=1$). Therefore, the requirement to access $E_{n}$ makes the alignment of synergy with autonomy rather obscure. This obscurity stems from whether to consider the requirement of accessing $E_{n}$ as a weakening argument for the system’s autonomy or not. The requirement of only accessing $S_{n}$ could be considered a superiority.

On the one hand, this confusion on the role of synergistic information in autonomy stresses the importance of PID. Since PID gives the opportunity to either consider CI or, if needed, discard it and rely on UI $\left(S_{n}\right)$, averting confusion that synergy poses. On the other hand, we speculate that autonomy is more often about the exclusivity of the system’s information rather than a requirement of accessing the agent’s internal state. This exclusivity is well captured by the unique information UI$\left(S_{n}\right)$, suggesting that it might be a more suitable measure for autonomy. Nonetheless, we would rather leave this matter on which $A_{0}$ or $\mathrm{UI}(S_{n})$ is a more suitable measure of autonomy as an open discussion that requires a more thorough inspection.

#### 5.2.2. PID Clarifies Autonomy When Agents Drive Their Environments

When the agent is driving the environment, Bertschinger et al. [6] suggested using $A^{*}=\mathrm{MI}(S_{n+1}:S_{n})$ to measure autonomy. The measure $A_{0}$ could be considered unsuitable because $A_{0}$ decays whenever the agent gains more control of its environment. The decay of $A_{0}$ might result from a UI $\left(S_{n}\right)$ decrease. This means that:
Autonomy measure $A^{*}=\mathrm{UI}(S_{n})+\mathrm{SI}$ will fluctuate w.r.t. training episodes; The only term accounting for autonomy is SI, given the argument by Bertschinger et al. [6] that $A_{0}$ eventually should decay to zero.

Therefore, it seems that the shared information, which reflects the coherence of the agent with the environment, could be a more suitable measure of autonomy than $A^{*}$. However, deciding on the suitability of $A^{*}$ or SI for capturing autonomy is another avenue to be investigated on more solid theoretical ground.

### 5.3. Relation to Previous Literature

Zhao et al. [14] defined an intrinsic reward based on mutual information between the agent and the environment to reward an RL agent. A more refined approach was taken in the current work by decomposing the mutual information using PID. However, we did not use the obtained values as an intrinsic reward but for characterising the agent’s behaviour.

Similar to our work, Zhao et al. [14] clearly separated the agent and the environment. They considered the setting as a Markov decision process (MDP) and assumed that the transition probabilities are known. This setting is similar to the grid environment setting described in Section 3.1 of the current study.

Seth [37] used a different approach to quantify autonomy. His approach was based on Granger causality, which allowed him to use an autoregressive model. The advantage of using regression is that it simplifies the estimation of probabilities compared to the approach by Bertschinger et al. [6]. Despite the difficulties in estimation pointed out by Seth [37], our results show interesting relationships between the information-theoretic quantities and the agent’s success.

Another information-theoretic quantity that is closely related to mutual information and has been used in RL is channel capacity. While mutual information is symmetric, channel capacity differentiates between the input and the output, denoted by *X* and *Y*, respectively. The channel is characterised by probabilities P(Y=y∣X=x) for possible values of *x* and *y*. Channel capacity is the supremum of mutual information over the possible distributions of input *X*.

Klyubin et al. [7] defined empowerment as the channel capacity with the agent’s action as the input and agent’s sensory input at a later time step as the output. Thus, empowerment quantifies the maximum amount of information that the agent could transmit through its actions into its sensory input. Since this transmission of information goes through the environment, one could perceive empowerment as the amount of control the agent has over the environment.

Jung et al. [8] generalise empowerment to continuous states and, unlike previous studies, they consider the case where transition probabilities are unknown. Their approach relies on Monte Carlo estimators, which can require a large amount of data to obtain accurate solutions. A more scalable approach is proposed by Mohamed and Rezende [9] using variational information maximisation.

We did not optimise mutual information as required for calculating empowerment, in the current work. Instead, we monitored the changes of different information-theoretic measures of the agent while the agent was trained using standard RL methods. Directly optimising a measure related to NTIC was conducted by Bertschinger et al. [6,38]. They chose the transition matrix using simulated annealing optimisation procedure and, thus, there was no direct link to standard RL.

According to Bertschinger et al. [38], maximising the mutual information between the agent’s action and its sensory input (as empowerment) can lead to informational closure. This means that the information flow from the environment into the agent, characterised by conditional mutual information MI$\left(S_{n+1}:E_{n}|S_{n}\right)$, tends to zero. This becomes trivial if the agent’s state does not contain information about the environment, that is, $M(S_{n+1}:E_{n})=0$. In the non-trivial case, NTIC, discussed in Section 2.1.3, quantifies the amount of closure when there is closure.

### 5.4. Limitations

One limitation of this work is that the standard RL frameworks like MDP and POMDP do not directly fit into the information-theoretic framework. The interactions between the agent and the state are slightly different from those depicted in Figure 1. In POMDP, the observation can also depend directly on action. The information-theoretic framework could always achieve this by sending the action through the environment’s state. Another difference is that in MDP and POMDP, the observation should go from $E_{n}$ to $S_{n}$ and not to $E_{n+1}$. The effect of this should be minor if the consecutive state changes are not close in time. We note that even if there are slight differences in the interactions, it is still possible to calculate the measures $A_{0}$, $A^{*}$, and NTIC using the given formulas; however, one has to be more careful in their interpretation. A possible solution to this is to separate the RL framework used in training from the information-theoretic framework used in the behaviour analysis, as has been done in this work.

Another limitation that could hinder the interpretation of the measures is when the underlying Markov chain is periodic. For an aperiodic Markov chain, we can interpret the probabilities $\mu (s^{\prime },s,e)$ introduced in Section 2.1.4 as the limiting distribution of the Markov chain. This interpretation essentially means that we consider the measures as the limiting values obtained if the Markov chain runs infinitely.

Finally, we note that we limited our work to discrete random variables. In the repeating-pattern environment, we had to bin the agent’s memory to have a discrete variable. According to Bertschinger et al. [6], the autonomy measures and NTIC can be generalised to continuous random variables. However, there are currently no convenient estimators for PID in the continuous case.

Pakman et al. [39] recently introduced an extension of BROJA to continuous random variables based on copula parameterised optimisation. However, their estimator can only handle variables of one dimension, which is insufficient for the repeating-pattern environment. In addition, Schick-Poland et al. [40] provided a measure-theoretic generalisation of SxPID definition that, in principle, handles mixed continuous-discrete random variables. Despite the mathematical rigour of the introduced measure, an estimator is still missing for the measure, prohibiting its usage. Therefore, due to the prematurity of the two continuous estimation methods, we resorted to using the discrete estimators of BROJA and SxPID.

### 5.5. Future Directions

This work focused on monitoring autonomy via PID-based measures. We have seen that PID-based measures aligned well and were robust in indicating the emergence of autonomy when it is expected. These findings pave the way to investigate further whether these PID measures constitute a necessary driving factor for the emergence of autonomy by using them, for instance, as intrinsic rewards.

Moreover, due to the generality of PID, it is a candidate to capture meta-learning concepts in general, and it is not restricted only to autonomy per se. Therefore, another line of research to pursue is developing PID-based cost functions (e.g., intrinsic ones) that motivate certain meta-learning aspects in addition to extrinsic cost functions. For instance, classical information-theoretic functionals have already been used to formulate intrinsic cost functions yielding improvements in performance [12,14]. In addition, this PID-based cost function could also be useful in multi-agent learning where it incentivises agents to learn specialisation, cooperation, or competition [12,41].

## 6. Conclusions

This work revolves around quantifying autonomy and several information-theoretic functionals in RL environments. To monitor autonomy and the internalisation of the environment by RL agents, we introduced an algorithm for calculating the measures $A_{0}$, $A^{*}$, and NTIC in the limiting process of time step approaching infinity. The introduced algorithm and techniques should unlock further practical applications. In addition to using the autonomy measures $A_{0}$, $A^{*}$, and NTIC suggested by Bertschinger et al. [6], we obtained fine-grained decompositions of these measures via PID, a recent extension of information theory.

We monitored autonomy and NTIC in various environments. These experiments showed that the autonomy measures aligned with the intuitive understanding of autonomy. Moreover, the PID fine-graining of $A_{0}$, $A^{*}$, and NTIC gave additional insights into understanding the behaviour of these measures in various environments.

PID can also be used to quantify how much an agent relies on the environment and how much it relies on its own internal state. Our perturbation experiments explored this reliance and gave some empirical evidence that these measures can be useful in practice.

Finally, we hope this work illustrates an example of utilising the PID framework to quantify abstract concepts in assessing and guiding learning algorithms.

## Figures and Tables

**Figure 1 entropy-24-00401-f001:**
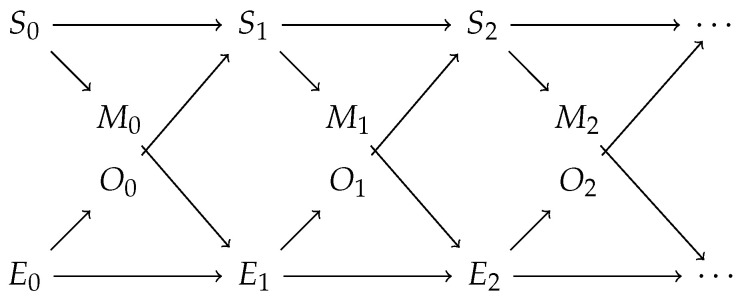
Interactions between the agent and the environment.

**Figure 2 entropy-24-00401-f002:**
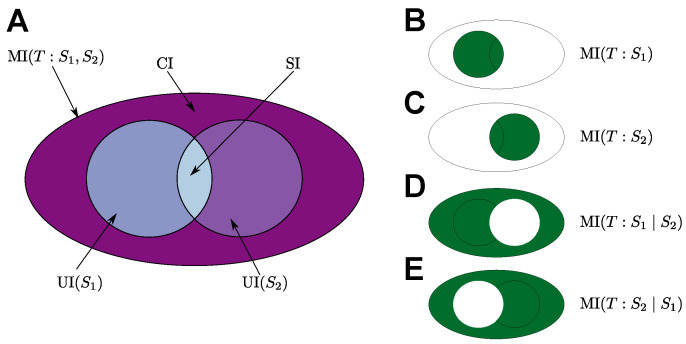
Partial information decomposition (PID) of mutual information partitions MI$\left(T:S_{1},S_{2}\right)$ into four information contributions of the sources $S_{1}$ and $S_{2}$ about the target *T* (depicted in **A**). In addition, this PID entails partitioning any information these sources have about the target, such as MI$\left(T:S_{1}\right)$ (depicted in **B**), MI$\left(T:S_{2}\right)$ (depicted in **C**), MI$\left(T:S_{1}|S_{2}\right)$ (depicted in **D**), and MI$\left(T:S_{2}|S_{1}\right)$ (depicted in **E**).

**Figure 3 entropy-24-00401-f003:**
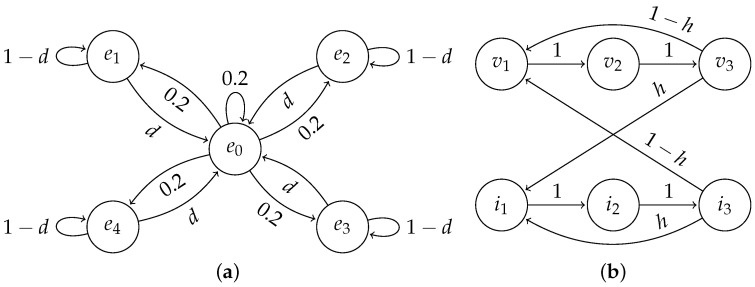
State transition diagrams of the environment’s state. (**a**) Grid environment; (**b**) Repeating-pattern environment.

**Figure 4 entropy-24-00401-f004:**
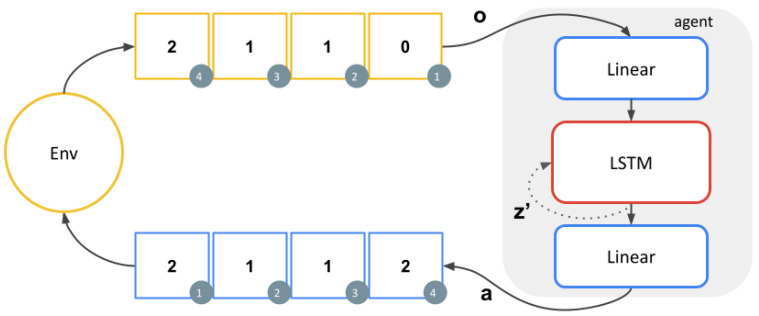
Illustration of the repeating-pattern environment with pattern (2,1,1). The current value of the pattern is hidden with some probability *h* by producing an observation of 0. The agent needs to use its memory $z^{\prime }$ to internalise the pattern sequence so it can maximise the reward even when the pattern is invisible.

**Figure 5 entropy-24-00401-f005:**
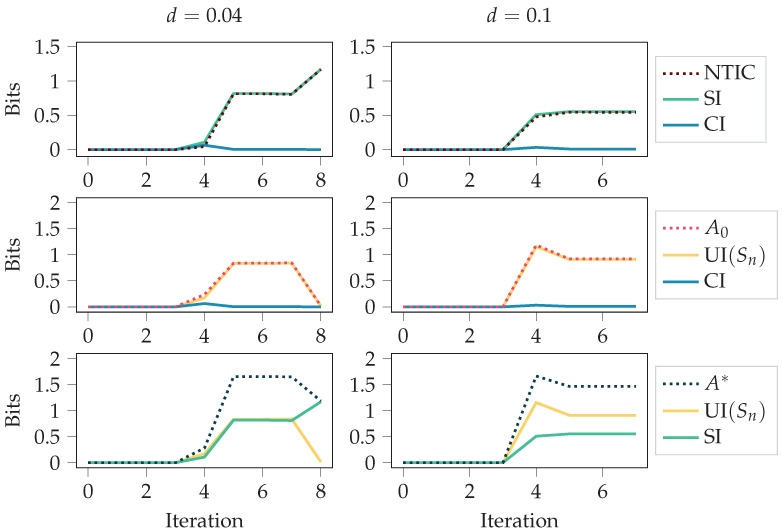
Shared information (SI) and non-trivial informational closure (NTIC) increase as the agent gains more control over its environment. The figure depicts autonomy measures and NTIC for the agents trained with policy iteration in the grid environment for different values of the food disappearing probability *d*. The figure also depicts synergistic information (CI) and unique information (UI $\left(S_{n}\right)$). See Figure A1 for the same quantities calculated with the SxPID estimator instead of BROJA.

**Figure 6 entropy-24-00401-f006:**
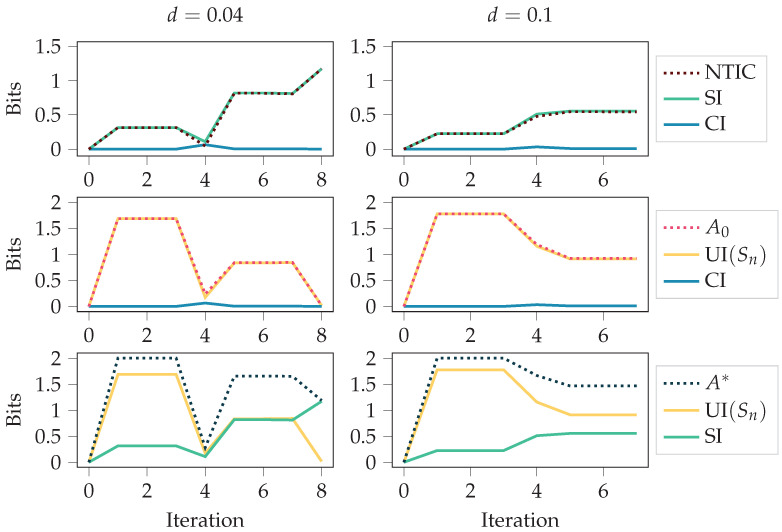
This figure is similar to Figure 5. The only difference is that the initial states $E_{0}$ and $S_{0}$ are chosen uniformly at random instead of having fixed initial states. The figure depicts autonomy measures and NTIC for the agents trained with policy iteration in the grid environment for different values of the food disappearing probability *d*. See Figure A2 for the same quantities calculated with the SxPID estimator instead of BROJA.

**Figure 7 entropy-24-00401-f007:**
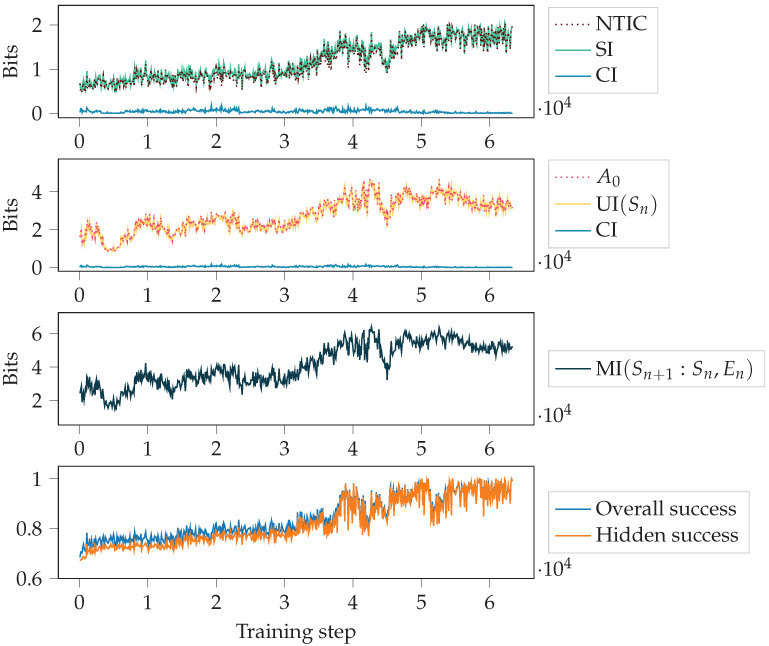
Autonomy and NTIC, together with their decomposition, the total mutual information MI$\left(S_{n+1}:S_{n},E_{n}\right)$, and the agent’s success throughout training. Here, h=0.9, and we consider one random initialisation of the agent.

**Figure 8 entropy-24-00401-f008:**
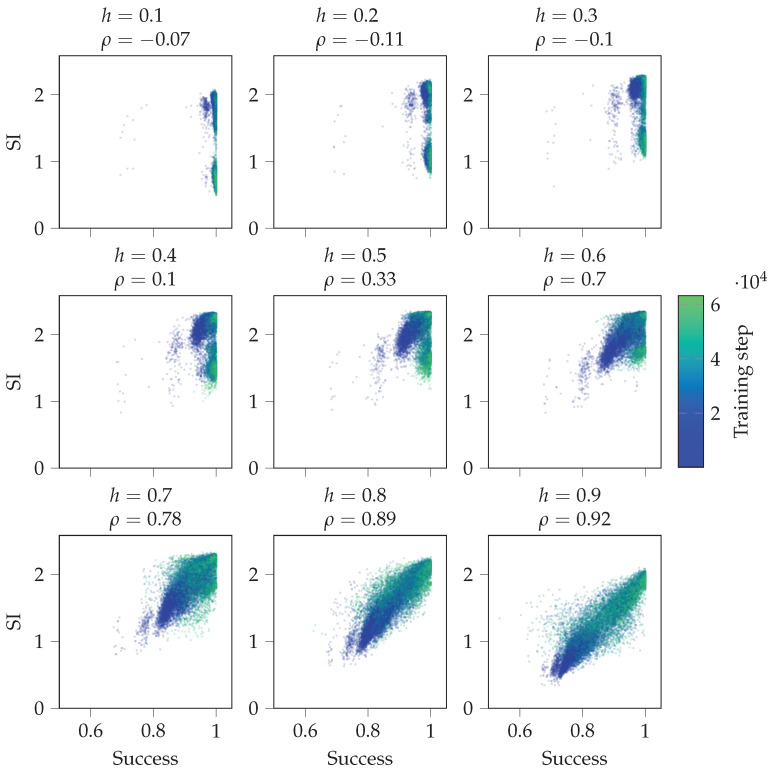
Shared information throughout training correlates with the success when the agent has an incentive to use its memory. The figure depicts scatterplots between overall success and shared information over the training process. Here, ρ denotes the Pearson correlation coefficient, and the data from all 30 random initialisations of the agent are included.

**Figure 9 entropy-24-00401-f009:**
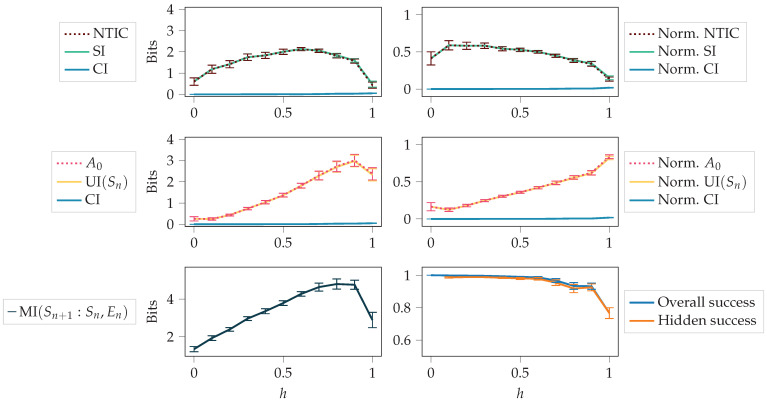
Unique information and $A_{0}$ should increase when an autonomous behaviour of the agent is the key to better performance. The figure depicts autonomy and NTIC, together with their decomposition, the total mutual information MI $\left(S_{n+1}:S_{n},E_{n}\right)$, and the agent’s success at the end of training. Here, the values are averages over the 30 random initialisations. Error bars show the 95% confidence interval (assuming the mean is normally distributed, which is approximately fulfilled due to the central limit theorem).

**Figure 10 entropy-24-00401-f010:**
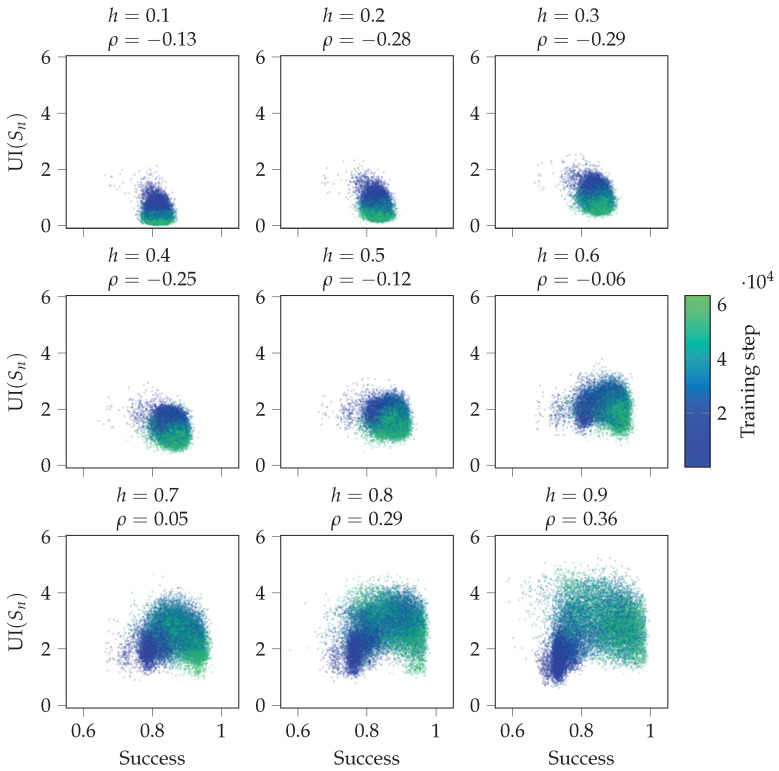
Scatterplots between overall success in the observed perturbation environment and unique information UI$\left(S_{n}\right)$ in the original environment over the training process. Here, *h* is the same in the original and perturbed environment, ρ denotes the Pearson correlation coefficient, and the data from all 30 random initialisations of the agent are included.

## Data Availability

The data generated and presented in this study are openly available here: https://github.com/antiingel/RL-agent-autonomy (accessed on 25 February 2022).

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
