# Peer review of "Quantifying Reinforcement-Learning Agent’s Autonomy, Reliance on Memory and Internalisation of the Environment"

_entropy, 2022, doi:10.3390/e24030401_

Round 1

Reviewer 1 Report

This is a very interesting paper that should definitely be published in Entropy.

The presentation, however, has to be improved, and in fact, the paper needs to be systematically rewritten. For instance, the information measures in the RL part are evaluated without telling the reader what the random variables are. That is revealed only much later. For a systematic presentation, one should start with the autonomy measures and PID, and then discuss the RL experiments as an application. That would make the paper logically coherent. That is, the authors have to reverse the presentation.

Also, for instance, why do you denote the autonomy measure by A_m when in the end you only consider the case m=0. Thus, it should be called A_0 if you want to keep the notation of Bertschinger et al.

Author Response

We thank the referees for their thoughtful comments and the time they devoted to assessing our work. We find that the issues raised in the reviews helped us to improve the manuscript and refine the presentation of our results.

Reviewer’s comment: The presentation, however, has to be improved, and in fact, the paper needs to be systematically rewritten. For instance, the information measures in the RL part are evaluated without telling the reader what the random variables are. That is revealed only much later. For a systematic presentation, one should start with the autonomy measures and PID, and then discuss the RL experiments as an application. That would make the paper logically coherent. That is, the authors have to reverse the presentation.

Response: The reason we had methods at the end was that at the time of writing, the journal formatting requirements explicitly asked for methods to be at the end. Recently the requirements have changed, and we can move methods to the beginning. We agree that it is easier to follow the writing this way, and this presentation is how we originally planned it. In short, we have now moved methods to the beginning (with one subsection moved to the appendix).

Reviewer’s comment: Also, for instance, why do you denote the autonomy measure by A_m when in the end you only consider the case m=0. Thus, it should be called A_0 if you want to keep the notation of Bertschinger et al.

Response: We have now changed A_m to A_0 in all places except the definition.

Reviewer 2 Report

This paper uses simulations of artificial reinforcement learning agents to study how measures of autonomy change as a function of the environment and training conditions. Th paper shows the evolution of these measures across training and their relationship to the agent's success.

General comments:

I struggled to understand why the motivation behind the simulations. In section 2 the reader is introduced to different environments without understanding yet why these are interesting or relevant to the broader problem. Section 3 introduces perturbations but again it's not clear what purpose they serve.

More generally, I struggled to understand what new insight is being offered by the paper. As I understand it, the idea that autonomy is driven by both shared information and unique information is definitional, and the fact that these different types of information are driven by different environmental parameters is also (I think?) definitional. Consequently, I fail to understand what non-trivial conclusion follows from the first sentence of the Discussion: "The results and the theoretical analysis suggest that when the agent affects the environment (grid environment experiments), autonomy manifests in the shared information between the internal state and the environment. Whereas, when the agent is solely driven by the environment (repeating-pattern experiments), autonomy could manifest in the unique information of the internal state." I apologize if I've missed something critical here.

Are there practical implications of this work? I couldn't really discern whether there was something here that would facilitate the construction of better RL agents. Or is this a purely conceptual exercise? The authors suggest a few ideas in section 4.5 but I would have liked to see something actually implemented.

I found the paper confusingly written. It introduces notation and concepts (e.g., O_n, NTIC) without explaining them first (instead referring to section 5). So it is hard to follow the interpretation of various measures throughout the methods and results. For example, the authors refer to shared information repeatedly, but it's only explained at the very end of the paper (and in a somewhat oblique way) what variables the shared information is defined over, a crucial detail.

Specific comments:

p. 3: The definition of state is confusing. Usually in a grid world, state would mean the agent's location, but the paper says that the state is the food location. I don't see how one can understand the agent's policy without including agent location as part of the state representation. Or I guess it's possible that "environment state" is different from "agent state" but this distinction isn't made clear in the paper. There's also "internal state" which makes things even more confusing. There is a notion of state that is relevant for the RL problem (what defines the Markov property in an MDP), and then there is another  notion of state that is relevant to the information-theoretic analysis. This needs to be clarified.

The authors should be aware that there are problems with simple histogram methods for estimating mutual information. See for example Paninski (2003) and Archer et al. (2013).

Fig 7: standard error bars or confidence intervals are more conventional than plotting standard deviation.

p. 12: "Correlation between shared information and success hints that shared information could be used as a generalised measure of success." I don't understand this. Why would we need another measure of success apart from just measuring success directly?

p. 13: "Pid" -> "PID"

Author Response

We thank the referees for their thoughtful comments and the time they devoted to assessing our work. We find that the issues raised in the reviews helped us to improve the manuscript and refine the presentation of our results.

First, we mention larger structural changes to the writing. We have now moved the methods section to the beginning (with one subsection moved to the appendix). The reason we had methods at the end was that at the time of writing, the journal formatting requirements explicitly asked for methods to be at the end. Recently the requirements have changed, and we could move methods to the beginning. This change helps to avoid confusion for readers not familiar with all of the used measures. Now we respond to each of the reviewer’s comments, multiple of which seem to stem simply from the location of the methods section.

All the line numbers, figure numbers and section numbers in the following refer to the first version of the manuscript.

Reviewer’s comment: I struggled to understand why the motivation behind the simulations. In section 2 the reader is introduced to different environments without understanding yet why these are interesting or relevant to the broader problem. Section 3 introduces perturbations but again it's not clear what purpose they serve.

Response: Motivation behind the simulations was first discussed in the Introduction in lines 59-63. Therein lines 62-63 mention the role of perturbations. Further, perturbations were discussed in lines 135-136 and in lines 205-207 (preceding section 3). More details were given about the perturbation in lines 300-304 (in section 3). We are not sure what aspect of motivation for perturbations is unclear at this point. We now added some more details to the beginning of the Experiment Setup section and moved the details of perturbation experiments from the Results section to the Experiment Setup section.

The motivation for using two different environments was discussed in the Experiment Setup in lines 69-76 (one environment provides a theoretically tractable case in which estimating probabilities is not required while the other environment better reflects a practical situation, using deep reinforcement learning (RL) and requires estimating transition probabilities). We have now added additional details about their differences. In particular, the measures A_m and A^* by Bertschinger et al. (2008) are suitable for measuring autonomy in different situations. Our repeating-pattern environment corresponds to the case where the environment drives the agent, and thus, A_m is the suitable measure. Grid environment allows the agent to affect the environment, in which case A^* could be considered more suitable. We now also recall the motivations for the environments in the corresponding Results sections.

The repeating-pattern environment gives a toy example for applying the analysis to deep RL. It provides a simple setting to test whether the agent relies more on its memory (the internal state in this case) or on the observation through the use of perturbations. If the agent is successful despite observations being perturbed, it is relying more on its memory. If the agent is successful despite the pattern being perturbed, it relies more on the observations. This was discussed in section 3.2.3.

The idea for grid-environment originates from Bertschinger et al. (2008), where some theoretical analysis about grid-like environments was already performed (we have now added a reference to this discussion). This environment gives a theoretically tractable case, and we demonstrated a technique for defining an internal state for the agent if it is not readily available through the training method (section 5.2 and 2.1.3). 

Reviewer’s comment: More generally, I struggled to understand what new insight is being offered by the paper. As I understand it, the idea that autonomy is driven by both shared information and unique information is definitional, and the fact that these different types of information are driven by different environmental parameters is also (I think?) definitional.

Response: First, we note that our paper introduces an algorithm that makes it possible to calculate the measures introduced by Bertschinger et al. (2008) in practice for RL agents (mentioned in lines 673-675, for example). Bertschinger et al. only optimised a measure using simulated annealing, and there was no direct relation to RL (as mentioned in lines 427-430). Thus we are taking the first steps to make the measures practically applicable in RL. We also demonstrate a technique for defining an internal state for the agent if one is not readily available. Since defining the underlying environment’s state and agent’s state is necessary for applying the method, this technique should also unlock further practical applications. Working out the theoretical details is a necessary precursor for practical applications. We have now added some additional details about it to the conclusions section.

Now about the definitions - indeed, the fact that the autonomy measures A_m and A^* introduced by Bertschinger et al. (2008) can be decomposed into PID terms (as discussed in section 5.4) follows directly from the definitions of A_m, A^* and PID. However, how the measures are changing throughout the training and what is causing the changes (with also the total mutual information changing) is a more complex question. Proposing to use PID to tackle this problem is one contribution of our work. This simplifies the analysis of the original measures and gives additional insights into why they are changing in this particular way. For example, using PID, we discovered that CI is close to zero throughout the whole training procedure in the repeating-pattern environment. 

Reviewer’s comment: Consequently, I fail to understand what non-trivial conclusion follows from the first sentence of the Discussion: "The results and the theoretical analysis suggest that when the agent affects the environment (grid environment experiments), autonomy manifests in the shared information between the internal state and the environment. Whereas, when the agent is solely driven by the environment (repeating-pattern experiments), autonomy could manifest in the unique information of the internal state." I apologize if I've missed something critical here.

Response: This is not a trivial consequence from the definitions since it is stated that the results and theoretical analysis suggest that only one of the two terms from the decomposition characterises autonomy (sections 4.2.1 and 4.2.2 gave a more in-depth discussion).

Reviewer’s comment: Are there practical implications of this work? I couldn't really discern whether there was something here that would facilitate the construction of better RL agents. Or is this a purely conceptual exercise? The authors suggest a few ideas in section 4.5 but I would have liked to see something actually implemented.

Response: Lines 298-300 stated that the agent’s strategy is usually treated as a black box, and using information-theoretic quantities could help to characterise the agent’s behaviour. As an example, we consider the perturbation experiments. We are using perturbed environments to directly evaluate (by calculating success in the perturbed environment) if the agent is relying more on its memory or the observations. On the other hand, we can estimate how much the agent is using its memory/observations to some degree through calculating corresponding PID terms (unique information in the chosen setting) in the original environment. Experiments suggest that higher dependency on memory indeed results in higher UI in the original environment. Thus, through information-theoretic measures, we have obtained some knowledge about how the agent would behave in the perturbed environment, and hence, on what it is relying on to make the decisions. We consider this to be a potentially practically useful implication for which we implemented experiments. We have now improved our writing to highlight these points better.

We consider this paper as an initial demonstration of the usefulness of PID-based measures of autonomy in the context of RL frameworks to the community of machine learning, which has little to no knowledge of PID. Thus, focusing the work on simple, understandable experiments motivates the community to adopt such an approach. Future work could follow that focuses on scaling the analysis to various RL environments of interest.

Reviewer’s comment: I found the paper confusingly written. It introduces notation and concepts (e.g., O_n, NTIC) without explaining them first (instead referring to section 5). So it is hard to follow the interpretation of various measures throughout the methods and results. For example, the authors refer to shared information repeatedly, but it's only explained at the very end of the paper (and in a somewhat oblique way) what variables the shared information is defined over, a crucial detail.

Response: We note that Materials and Methods was the last section due to the formatting requirements of the journal. We have now moved it to the beginning, as we already mentioned. This should make the writing clearer. However, we do not agree that the details given in the Materials and Methods are given “in a somewhat oblique way”. PID is introduced generally in section 5.3, followed by its application to our case in section 5.4, in which the source and target variables used in PID decomposition are clearly stated. 

We note that in the previous version of the manuscript, each first occurrence of a notation that was in detail described in the Methods section at the end, there was a reference to the corresponding section so that readers not familiar with the concept could read this section first.

Reviewer’s comment: p. 3: The definition of state is confusing. Usually in a grid world, state would mean the agent's location, but the paper says that the state is the food location. I don't see how one can understand the agent's policy without including agent location as part of the state representation. Or I guess it's possible that "environment state" is different from "agent state" but this distinction isn't made clear in the paper. There's also "internal state" which makes things even more confusing. There is a notion of state that is relevant for the RL problem (what defines the Markov property in an MDP), and then there is another  notion of state that is relevant to the information-theoretic analysis. This needs to be clarified.

Response: There is no one single definition of a state since we are using multiple different notions of state, and we are doing our best to differentiate between these notions. Most of the comment seems not to be a confusion but rather an understanding of different notions of states we are using. With methods at the beginning, there should be less confusion now. We follow up with some clarifying comments.

First, there are states for the information-theoretic framework: environment’s state and agent’s (internal) state. The distinction between these two was already made clear in the Introduction. Throughout the paper, we always specify if we mean the environment's state or the agent’s state to avoid confusion. These states were generally defined in section 5.1.2 (the same way as Bertschinger et al. (2008)).

How the general framework is applied to the specific environments was discussed in the corresponding sections. For specific cases, the states were defined in the corresponding sections (2.1.1, 2.1.3, 2.2.1, 2.2.3), each titled either “States of the Environment” or “Internal States of the Agent” to make it clear what it is discussing.

Then there is another notion of state used in the training frameworks. Section 2.1.2 discusses MDP used in training and its states. A distinction is made between what is a state in terms of the information-theoretic framework and what is a state for MDP. We have now added an additional comment that the states in terms of training framework are different from the information-theoretic framework states. Lines 103-104 stated that MDP state contains the location; therefore, this information is available to the agent and should not have caused confusion.

Reviewer’s comment: The authors should be aware that there are problems with simple histogram methods for estimating mutual information. See for example Paninski (2003) and Archer et al. (2013).

Response: First, we note that for the grid environment, there is no need for estimating probabilities as the probabilities are known, and the computations are using exact probabilities. We give some comments about the repeating-pattern environment and estimating therein.

Paninski (2003) states that for a fixed number of bins (which is our case as we have simply fixed the number of memory bins), the convergence is guaranteed by classical results. Archer et al. (2013) simply refer to Paninski (2003) and another paper regarding estimation problems.

We are not claiming that our method is a good way for estimating measures in the continuous (or mixed continuous-discrete case). On the contrary, we claimed that we are working exclusively on discrete (actually even finite) cases (line 456, lines 499-500). This we achieved by simply discretising the memory and taking the discretised version as the definition for the agent’s state. 

The reason for considering only discrete cases was discussed in lines 456-468.

Reviewer’s comment: Fig 7: standard error bars or confidence intervals are more conventional than plotting standard deviation.

Response: We have now generated more data to calculate (approximate) confidence intervals and replaced standard deviations with confidence intervals. More data was needed to achieve approximate normality of the mean since the actual distribution was unknown. Since now there is more data, we also updated figures 4, 6, 8 and A1 accordingly.

Reviewer’s comment: p. 12: "Correlation between shared information and success hints that shared information could be used as a generalised measure of success." I don't understand this. Why would we need another measure of success apart from just measuring success directly?

Response: Indeed, being simply a measure of success does not really highlight the usefulness of the measure. We note, however, that our experiments show that SI can indeed be calculated in practice and in our experiments, it coincides with our intuitive expectations. Thus our results provide empirical evidence that SI (or other discussed measures) can be monitored to characterise the agent’s strategy. We have now changed the paragraph to focus more on the monitoring aspect. In future work, these measures could be considered as intrinsic rewards for the agent. 

Reviewer’s comment: p. 13: "Pid" -> "PID"

Response: We have now fixed the typo.

Reviewer 3 Report

In this paper authors examine changes of autonomy of an reinforcement learning agent during training. They use partial information decomposition framework and make some numerical tests for two environments. They analyse how much agent decisions rely on the internal memory and how much on the observations it makes.

Obtained results are interesting and provide some insight into reinforcement learning algorithms. I recommend publication, but have a minor comment. The symbol Theta that appears in line 111 should be defined or at least a reference should be given where it is defined.

Author Response

We thank the referees for their thoughtful comments and the time they devoted to assessing our work. We find that the issues raised in the reviews helped us to improve the manuscript and refine the presentation of our results.

First, we mention larger structural changes to the writing. We have now moved the methods section to the beginning (with one subsection moved to the appendix). The reason we had methods at the end was that at the time of writing, the journal formatting requirements explicitly asked for methods to be at the end. Recently the requirements have changed, and we could move methods to the beginning. This should make the writing more easily readable.

Reviewer’s comment: The symbol Theta that appears in line 111 should be defined or at least a reference should be given where it is defined.

Response: We have now restructured the sentence to make the meaning of theta clearer and added again a reference to the algorithm description, where it is defined.

Round 2

Reviewer 2 Report

I thank the authors for clarifying some aspects of the paper which were confusing to me. I have to admit that I still don't grasp in what way the contributions of this work are useful to central questions and challenges in reinforcement learning. However, I acknowledge that some researchers might find the measures developed here useful (clearly the authors do), so I don't want to create obstacles just because of my own point of view. My view is that the paper is not currently written to appeal to a broad audience, but we don't need every paper to be written in such a way.

Author Response

We have now added more references to the introduction to clarify how our work relates to the central questions and challenges in AI or RL; the rest of the text was improved in the previous revision.

In particular, we now added references to in-depth discussions about perturbation tolerance and the importance of autonomy in AI. These are widely discussed problems in AI, and there are definitely many researchers interested in these topics. These are complex problems and require step-by-step solutions. We tackle these problems in an information-theoretic setting, introducing an algorithm to monitor an RL agent's autonomy and a certain kind of perturbation tolerance in this setting. Being able to measure autonomy or perturbation tolerance is crucial to more advanced applications. For example, self-monitoring is considered a fundamental requirement for perturbation tolerance. We note that other characterising measures could also be monitored thanks to the generality of the information-theoretic setting we are using.